# A participatory approach to move towards a One Health surveillance system for anthrax in Burkina Faso

**Sougrenoma Désiré Nana**[1,2], **Raphaël Duboz**[1,3,4], **Potiandi Serge Diagbouga**[5,6], **Pascal Hendrikx**[7], **Marion Bordier**[1,3,8]*

1 UMR ASTRE, University of Montpellier, CIRAD, INRAE, Montpellier, France, 2 UMR ASTRE, CIRAD, Montpellier, France, 3 UMR ASTRE, CIRAD, Dakar, Senegal, 4 UMMISCO, IRD, Sorbonne University, Bondy, France, 5 Research Institute of Health Sciences, Ouagadougou, Burkina Faso, 6 Health Training and Research &Development, Ouagadougou, Burkina Faso, 7 High Council for Food, Agriculture and Rural Areas, Paris, France, 8 National Laboratory for Livestock and Veterinary Research, Senegalese Institute of Research in Agriculture, Dakar, Senegal

* marion.bordier@cirad.fr

**Data Availability Statement:** All relevant data to support the replication of this study are within the paper, except the name and contact details of

## Abstract

The One Health approach calls for collaboration across various sectors and different scales to improve understanding of complex health issues. Regarding epidemiological surveillance, this implies the development of integrated systems that link several surveillance components operating in different domains (human, domestic animals, environment) and involving several actor networks. However, surveillance continues to operate in a very compartmentalized way, with little interaction between sectoral institutions and with the community for the governance and operation of surveillance activities. This is partly explained by the insufficient consideration of the local context and the late involvement of national stakeholders when developing programmes that aimed at strengthening the integration of surveillance. In low- and middle-income countries in particular, there is a strong influence of external partners on the development of intersectoral programmes, including surveillance systems. In this context, we developed and implemented a participatory planning process to support stakeholders of the surveillance system of anthrax in Burkina Faso, in the definition of the One Health surveillance system they wish for and of the pathway to reach it. The workshop produced an action plan that reflects the views and perspectives of representatives of the different categories of stakeholders and beneficiaries of surveillance. In addition, the participation of stakeholders in this participatory co-construction process has also improved their knowledge and mutual understanding, fostering a climate of trust conducive to further collaboration for surveillance activities. However, the quality of the participation raises some questions over the results, and contextual factors may have influenced the process. This underlines the need to include a monitoring and evaluation plan in the process to assess its implementation and ability to produce One Health surveillance modalities that are appropriate, accepted and applied over the long term.

workshop participants which remain confidential for ethical reasons.

**Funding:** This work was funded in part by CIRAD, VetagroSup (the French institute for higher education and research in food, animal health, agricultural and environmental sciences) and the French Embassy in Burkina Faso.

**Competing interests:** The authors have declared that no competing interests exist.

## Introduction

The One Health approach promotes collaboration between sectors, professions, and disciplines to improve the health of humans, animals, and ecosystems on an equitable basis [1]. The application of this concept to epidemiological surveillance is expected to result in greater epidemiological and economic performance of surveillance systems. Indeed, it is expected to improve knowledge of health events and their management while reducing the operational costs of surveillance [2–6]. Less tangible benefits may also occur, such as the establishment of mutual understanding (i.e., understanding one another's thoughts, feelings, and perspectives) and trust between actors, conducive to further and deeper collaboration [4,7]. However, the operationalization of the principles of the One Health concept in the context of surveillance remains limited [6,8]. This is particularly the case in Burkina Faso, where, despite the establishment of an inter-ministerial platform to address health issues, and strong advocacy and substantial technical and financial support from international organizations and cooperation agencies, zoonosis surveillance remains very poorly integrated and collaborative [9].

Participatory processes were initially proposed in the 1980s as an alternative to more traditional approaches in which dominant stakeholders determine priorities and objectives according to their own interests and agendas [10]. The introduction of participatory processes thus represented an attempt to give a greater voice to categories of stakeholders usually excluded from decision-making [10]. Rowe and Frewer (2004) [11] define participation as "the practice of consulting and involving relevant stakeholders in the identification of priorities, decision-making and policy-making activities of organizations or institutions responsible for policy development". In recent decades, participation has grown significantly and is increasingly mobilized to involve all stakeholders in decision-making to address issues in a variety of fields, such as technology, the environment and health [12].

In this vast field of participation, participatory planning is a particular method that aims to establish a dialogue between various stakeholders to collectively define the actions to be implemented to achieve a common goal [12]. In this approach, it is postulated that engaging beneficiaries early in the planning process establishes a collective vision that can be more effectively realized. The use of a participatory planning process therefore emerged as a contribution to the operationalization of One Health surveillance in Burkina Faso, by offering surveillance stakeholders and beneficiaries a framework for the collective definition of the desired surveillance system, and the means to achieve it.

To this end, a participatory workshop was held with representatives of the different categories of surveillance stakeholders to collectively define such a shared vision, and then identify the changes, and associated actions, necessary to evolve from the current situation to the desired situation.

## Materials and methods

To conduct the participatory process, we developed a method inspired by the one proposed by Bordier et al., 2019 [13], which we implemented during a three-day workshop. We will first describe the general organization of the workshop, before detailing the various steps of the method, namely: (i) identification of a shared vision of the One Health surveillance system of anthrax in Burkina Faso; (ii) co-construction of a collective representation of the current situation of the surveillance system in place, in the form of a stakeholder diagram; (iii) identification of the changes needed to move from the current surveillance system to the desired surveillance system; and (iv) drafting of an action plan to achieve the vision of the future One Health surveillance system (Fig 1).

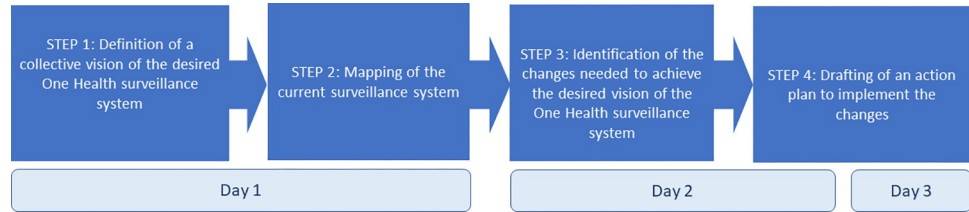

**Fig 1. The four steps in the participatory process for the co-construction of the One Health surveillance system of anthrax in Burkina Faso.**

## Workshop organization

In a co-construction process, the quality of the participants is crucial, as their representativeness has a very strong influence on the ability of the process to produce results that represent the vision and expectations of all categories of stakeholders. The workshop participants were selected based on a mapping of anthrax surveillance stakeholders in Burkina Faso conducted in a previous study [9]. The aim was to gather a representative of each of the actors involved in the various surveillance functions, while maintaining a group size that would be manageable within the framework of a participatory process. As a result, 16 institutional representatives were invited to the workshop. Table 1 summarizes the list of invited institutions. "Local authorities" refers to field actors working for ministries in charge of public health, animal health and wildlife conservation. In the case of this workshop, we chose an easily accessible meeting place, outside any sectoral ministry, so as not to give more weight to one sector than another. Invitations were issued by the Ministry of Higher Education, Research and Innovation.

Facilitation is also an important element in the success of a participatory process; the quality of facilitation determines the relevance of the results produced during the workshop. Facilitators need to be agile in leading discussions with participants from different backgrounds, using appropriate participative tools. Their role includes encouraging participants to clarify what they are saying when there is a risk of misinterpretation, and even to rephrase, if necessary, to ensure that the whole audience understands [14,15]. The co-construction workshop was facilitated by two researchers (one specializing in epidemiological surveillance and another in complex systems and support modelling) and a PhD student from Burkina Faso working on integrated surveillance of anthrax in the country. The process included an evaluation plan to assess the quality of the results produced, as well as the effects of the workshop on participants

**Table 1. List of invited institutions and number of participants.**

| Sector | Category of actors | Number of participants invited | Number of participants present | | |
|--------|--------------------|--------------------------------|--------------------------------|-----|-----|
| | | | **1st day** | **2nd day** | **3rd day** |
| Multisectoral | National authorities | 4 | 1 | 2 | 2 |
| Animal health | Central authorities | 1 | 1 | 1 | 1 |
| | Local authorities | 2 | 2 | 2 | 2 |
| | Central laboratory | 1 | 0 | 0 | 0 |
| | Regional laboratory | 1 | 1 | 1 | 1 |
| Human health | Central authorities | 1 | 1 | 1 | 1 |
| | Local authorities | 1 | 1 | 0 | 1 |
| Wildlife conservation | Central authorities | 1 | 1 | 1 | 1 |
| | Local authorities | 2 | 2 | 2 | 2 |
| International aid | International technical and financial partners | 2 | 2 | 1 | 1 |

and on the wider implementation context. Qualitative and semi-quantitative indicators were measured based on data collected before, during and after the workshop, using different methods (questionnaires, semi-structured interviews, workshop observation). This plan will not be presented here but will be the subject of another publication.

### Identification of a shared vision for a One Health surveillance system for anthrax in Burkina Faso

Before starting the discussions towards a shared vision of the desired surveillance system, the first step was to collectively agree on a definition of "integrated surveillance system" based on a One Health approach for the purposes of the exercise. Each participant was asked to write down a word or a short phrase which, in their opinion, defines or characterizes a One Health integrated surveillance system. The proposals made by the participants were then grouped together by theme to arrive at a common definition of One Health surveillance. The next step was to define a common vision of the One Health surveillance system over a 10-year timeframe. This seemed to be the most appropriate timeframe to enable participants to envision a functional One Health surveillance system in the context of Burkina Faso. Participants were encouraged to write a short narrative about their vision of the ideal system in terms of objectives, performance, organization and operation, and the role they see themselves playing in it. The participants took it in turns to read their notes, and a thematic grouping of the various characteristics of the desired system was drawn up as they went along by the co-facilitator (Fig 2).

### Co-construction of a collective representation of anthrax surveillance in Burkina Faso in the form of a stakeholder diagram

To enable participants to project the changes required to achieve the desired One Health surveillance system, it was necessary for them to begin by collectively defining a common representation of the current organization and operation of surveillance. To this end, they were supported in the co-construction of an actor diagram focusing on information flow, in which the roles and missions of all the actors involved in the surveillance system were represented, as well as their interactions. The aim was for participants to learn from each other and produce new knowledge, fostering the development of mutual understanding [15]. A preliminary actor diagram, drawn up in advance of the workshop based on the work of Nana et al. (2022) [9], was presented to the participants. Participants were invited to propose modifications after pointing out their place in the diagram to the whole audience. The facilitator made modifications as they went, after validating them with all the participants (Fig 3).

### Identification of the changes needed to achieve the desired vision

A "back casting" method was used to trace backwards from the vision of the future to the current situation. This method is notably used in anticipatory approach to help stakeholders define a roadmap for achieving sustainable and desirable futures at the scale of a given territory and in the context of a specific issue [16].

In this step, workshop participants were asked to identify the changes needed to move from the current situation to their vision of the future. The characteristics of the desired situation were first mentioned at one end of a timeline, then, for each of them, a description of the current situation was collectively defined and mentioned at the other end of the timeline. The facilitator then asked participants to record, by writing on a card, any changes needed to reach each desired characteristics, starting from the future and working backward to the present. Changes were expressed in terms "which actor needs to do what differently". The facilitator

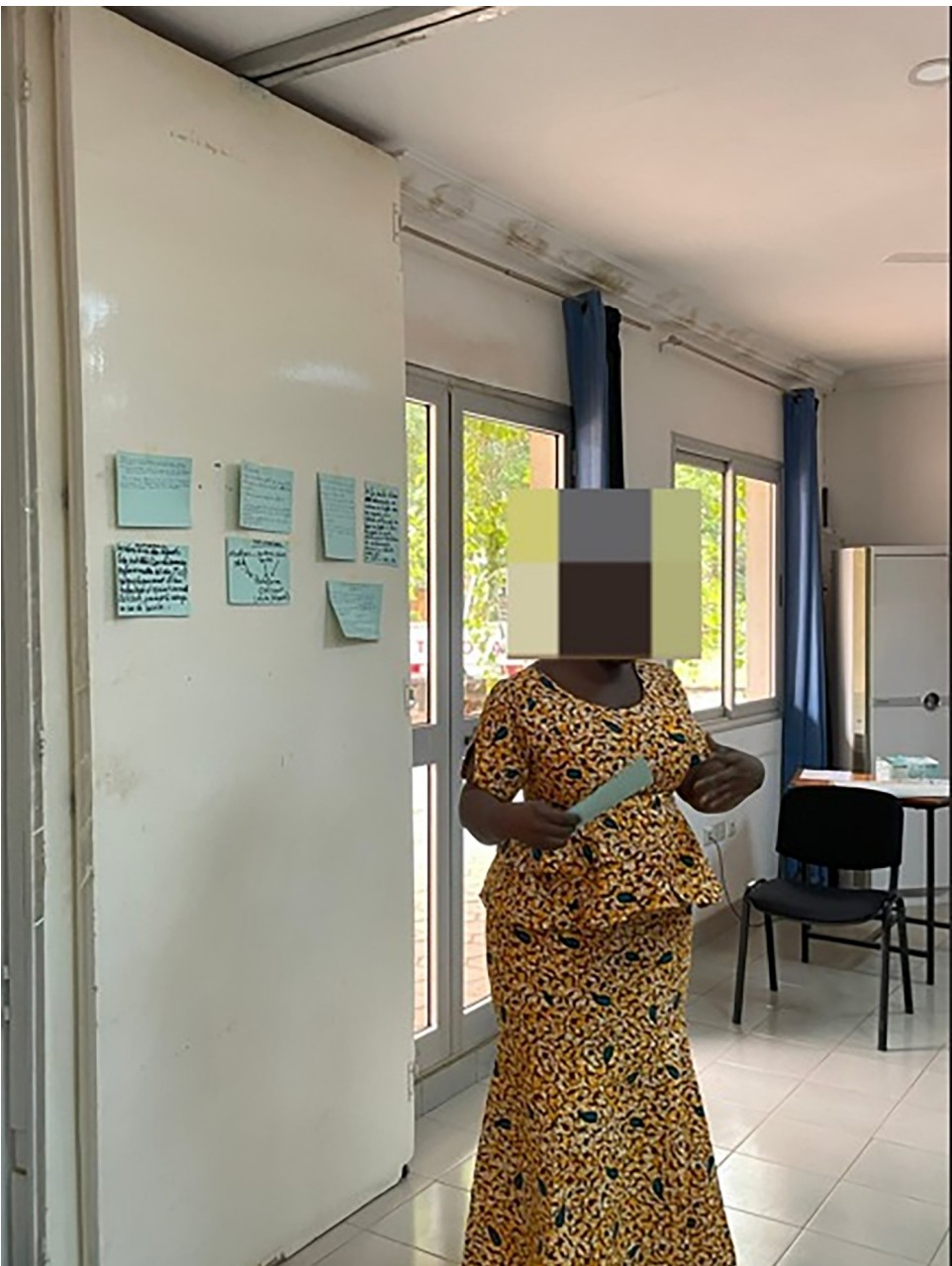

**Fig 2. Collective definition of the vision of the One Health surveillance system for anthrax desired in 10 years' time in Burkina Faso.**

asked participants to arrange the changes in chronological relation to each other until a path of changes was constructed, to move from the current situation to the desired situation (Fig 4). The participants then identified causal links between the various proposed changes.

### Design of an action plan to operationalize changes to achieve the desired vision

Once the changes needed to achieve the desired vision of the One Health surveillance system for anthrax had been identified, participants drafted an action plan to operationalize the

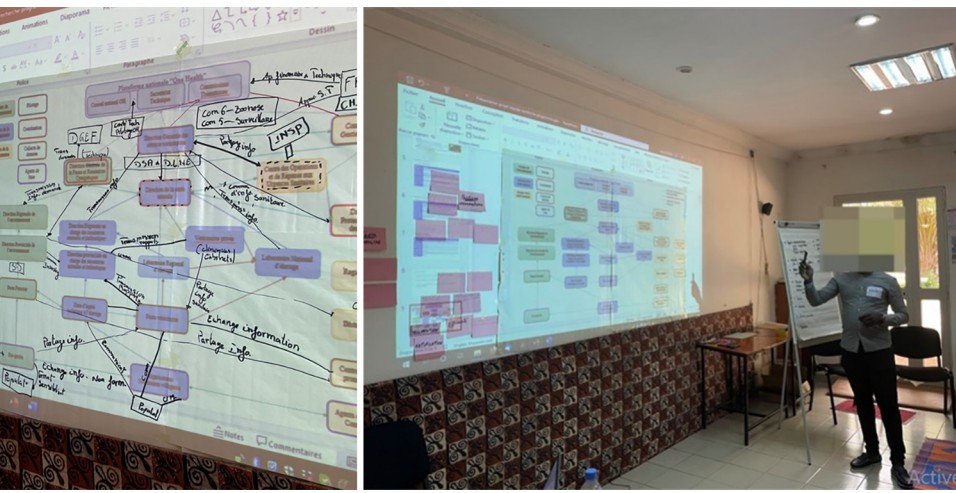

**Fig 3. Collective revision of the diagram of anthrax surveillance actors in Burkina Faso.**

changes identified. This reflective step was also an opportunity to identify levers and obstacles regarding those changes, which were also recorded on the pathway of changes. Once this consolidation work had been completed, the changes were considered one by one and participants asked to identify the necessary actions for their respective implementation, and to characterize them in terms of: beneficiaries; responsible; implementer; source of funding; possible levers and obstacles regarding implementation; date; and duration of implementation. Due to time constraints and the disruption of the workshop by external political events, the action plan could only be partially outlined.

## Ethics approval and consent to participation

This study was evaluated and validated by the ethics committee of the Ministry of Higher Education, Scientific Research and Innovation of Burkina Faso in March 2021 by deliberation n° 2021–07–161. A written informed consent to workshop participation was obtained from all

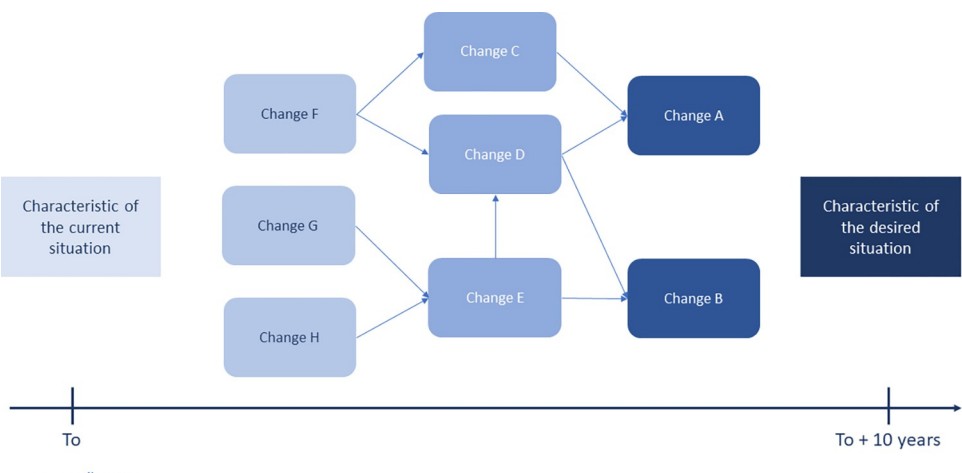

**Fig 4. Change pathway model used to construct the transition from the current to the desired situation of the One Health surveillance system for anthrax in Burkina Faso.**

participants. The study protocol complies with applicable guidelines. The recruitment period of participants started on 7 August 2022 with the selection of individuals to be invited and ended on 3 April 2023 with the completion of all interviews conducted to assess the effects of the co-construction process (not presented here).

## Results

The co-construction workshop took place from 28 to 30 September 2022 in Ouagadougou, Burkina Faso, in a neutral place, outside of any sectoral ministries. The number of participants varied throughout the co-construction workshop: 12 on the first and third days, and 11 on the second day. The number of participants on each day by sector and stakeholder category is shown in Table 1.

### A shared vision of One Health surveillance for anthrax in Burkina Faso

The discussion on the definition of integrated surveillance based on a One Health approach led to an agreed definition of the term. Workshop participants defined it as a coordinated and collaborative system between different sectors, enabling the sharing of data and information needed for effective risk management. Such a system requires the production of quality data and is based on the principle of transparency between all actors. The participants agreed on a definition of an integrated anthrax surveillance system within 10 years, based on three pillars: (i) notification and investigation of all cases of anthrax in Burkina Faso; (ii) proper circulation and use of the information generated by surveillance; and (iii) effective intersectoral governance of surveillance. These three pillars result from the thematic grouping based on the characteristics of the ideal system proposed by the participants, and are represented in Fig 5.

### Collective representation of the anthrax surveillance system in Burkina Faso in the form of an actor diagram

The revision of the actor diagram led to the addition and deletion of actors and interactions. All these changes were made with a high degree of consensus, except for the interactions linked to information sharing between field actors, which gave rise to differences of opinion. In fact, these exchanges are not formalized in an official document, and, as a result, actors from central authorities wanted them not to be represented. Local actors objected, arguing that these interactions existed and were often more effective in managing health events than those taking place at central level, which are formalized. A compromise was reached to distinguish

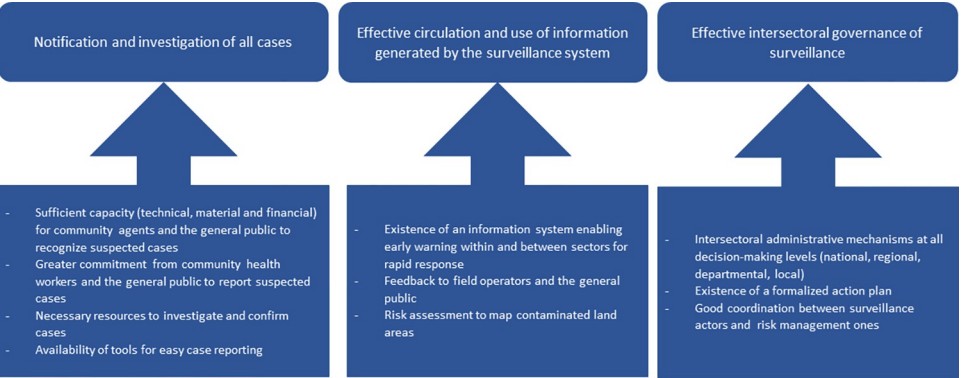

**Fig 5. Characteristics of the One Health surveillance system for anthrax desired in 10 years' time in Burkina Faso.**

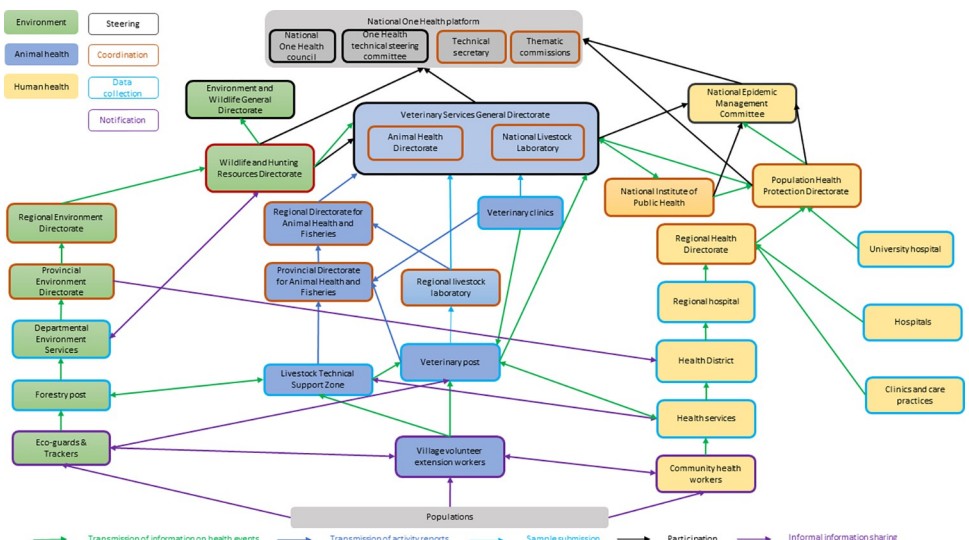

**Fig 6. Diagram of the actors in anthrax surveillance system in Burkina Faso.**

formalized and non-formalized interactions on the diagram. The diagram thus amended highlights the importance of informal collaboration in surveillance and is illustrated in Fig 6.

## Changes needed to achieve the desired vision

**Changes needed to achieve the notification and investigation of all cases.** The proposed changes had four specific objectives (Fig 7A and 7B). The first was to improve the capacity of local authorities to recognise suspected cases of anthrax (through posters, picture boxes and training) and to notify them (through forms and the deployment of electronic reporting tools). The second objective was to raise public awareness of the need to notify suspected cases. A third objective was to provide the resources needed to confirm suspected cases, in particular by setting up an emergency fund. The final objective was to improve notification procedures by deploying electronic reporting tools for community workers.

**Changes required for proper circulation and use of the information generated.** Changes for improving the circulation and use of the information have been proposed in three domains (Fig 8). To ensure that the information system enables early warning and rapid response, interoperable information systems should be developed and deployed across the country. The information must then be fed back to field agents and communities using the same channel as the one used for case notification. Finally, any contaminated land areas must be properly mapped based on an updated risk assessment and their coordinates communicated to farmers.

**Changes needed for effective cross-sectoral governance of surveillance.** Fig 9 illustrates the various changes and their implications for effective cross-sector governance of epidemiological surveillance. First, intersectoral coordination committees for zoonosis surveillance need to be established at sub-national level and the technical, and interpersonal skills of committee members enhanced. Next, a specific intersectoral strategic plan for surveillance must be formalized, regularly updated and integrated into the global intersectoral strategic plan for zoonoses. This integration would give greater visibility to the actions proposed for anthrax surveillance and improve the chances of obtaining funding for their implementation. Lastly, coordination must be improved between actors in charge of surveillance and those in charge of

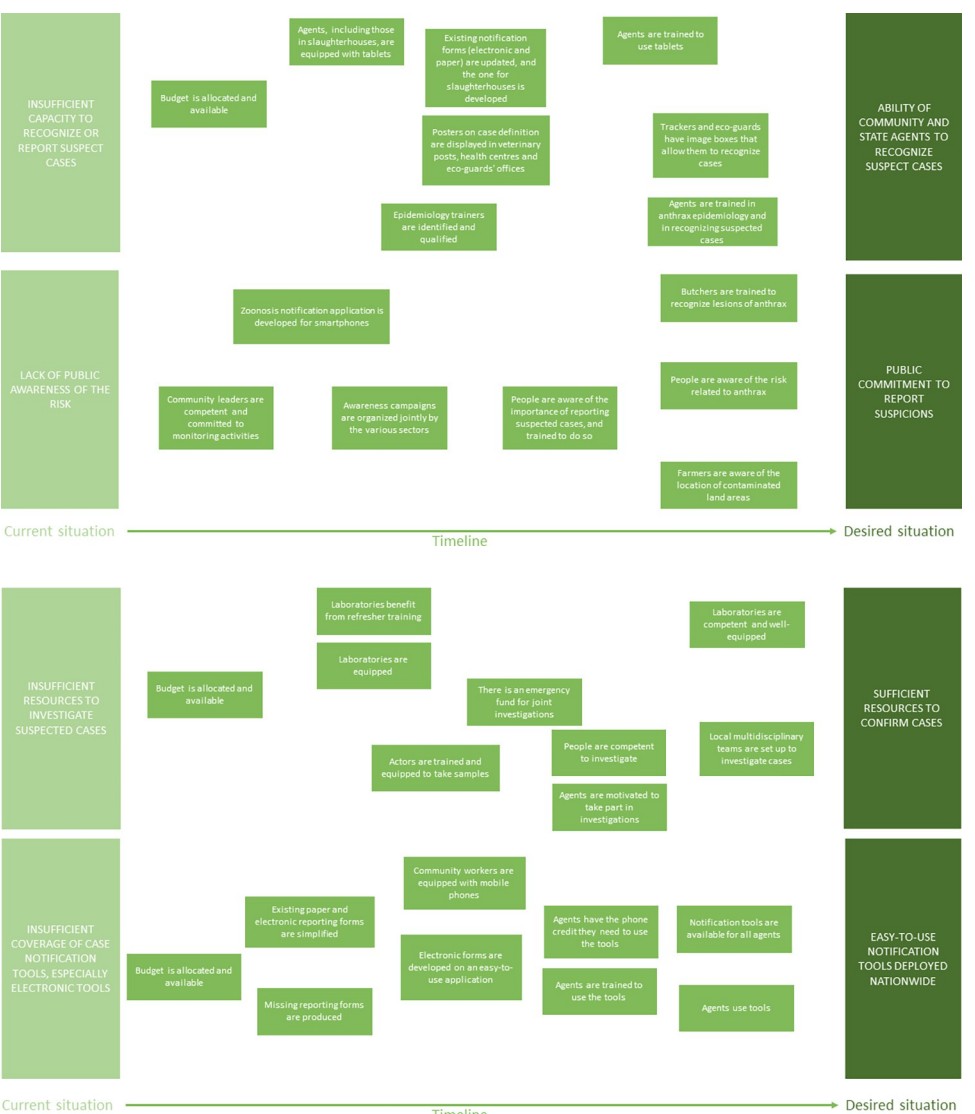

**Fig 7.** a and b Necessary changes identified by participants to ensure notification and investigation of all cases.

management, notably through information exchange within the local intersectoral committees for zoonoses surveillance.

**Draft action plan for implementation of the desired One Health surveillance system.** The action plan could only be completed for the first pillar, namely notification and investigation of all anthrax cases in Burkina Faso (Table 2). For this pillar, 13 changes were selected by the participants for inclusion in the action plan, and 37 actions were proposed to implement these changes. Nineteen actions were aimed at improving the technical capacities of surveillance actors, of which 15 aimed at fostering interaction between actors. Three actions were dedicated to mobilizing the financial resources needed to implement the entire plan. The participants identified levers for and barriers to the implementation of the actions formulated. The existing national One Health platform represents an appropriate governance framework for implementing the actions identified. However, the lack of political will to allocate more resources for surveillance, as well as political instability and insecurity in large parts of the country, were identified as potential barriers to the implementation of many of the plan's actions.

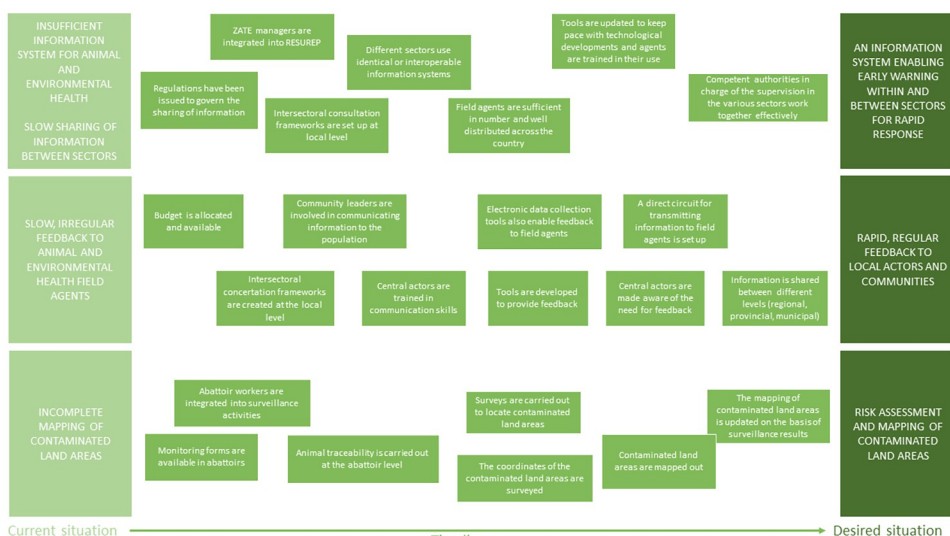

**Fig 8. Necessary changes identified by participants in the co-construction workshop to ensure proper circulation and use of the information generated (RESUREP: Réseau de surveillance épidémiologique (Epidemiological Surveillance System); ZATE: Zone d'Appui Technique à l'Elevage (Livestock Technical Support Zone).**

## Discussion

The co-construction process implemented in Burkina Faso proved fruitful in supporting stakeholders towards the implementation of a One Health surveillance system for anthrax. It led to collective representation of the current surveillance system and of the desired surveillance system, as well as the definition of changes and actions to move from one to the other. However, because of time constraints, the action plan was not finalised.

In the course of defining One Health surveillance, participants from the animal health and environmental sectors demonstrated a good understanding of what surveillance means in a One Health approach, and of the issues involved. Some participants from the human sector,

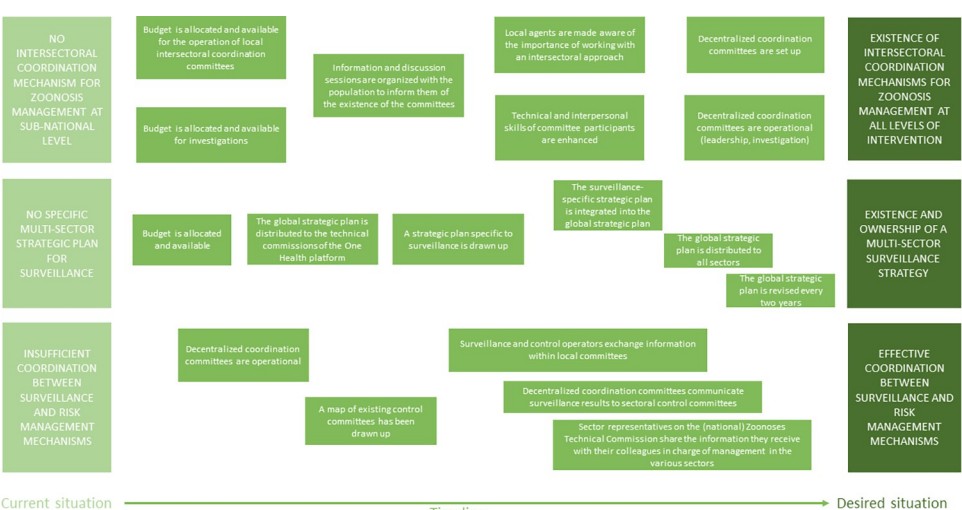

**Fig 9. Necessary changes identified by participants in the co-construction workshop to ensure effective cross-sector governance of surveillance.**

**Table 2. Action plan for the notification of all cases of anthrax in Burkina Faso.**

| CHANGES | | ACTION | | ACTION BENEFICIARIES | MANAGER | EXECUTANT | ORIGIN OF FUNDS |
|---|---|---|---|---|---|---|---|
| 1. | Budget is allocated and available | 1.1 | Draw up an action plan (development/validation workshops) | Key surveillance actors | TS—OHNP | Zoonoses commission (a core group of 10 people to draft 0) | TS-OHNP/TFP/Sectoral |
| | | 1.2 | Advocacy with the government and TFPs | Key surveillance actors | TS—OHNP | Zoonoses commission | TS-OHNP/TFP/Sectoral |
| | | 1.3 | Develop an annual staffing program | Key surveillance actors | TS—OHNP | TS—OHNP, technical directorates | TS-OHNP/TFP/Sectoral |
| 2. | Agents, including those in abattoirs, are equipped with tablets | 2.1 | Buy the tablets | Authorities (animal health and environment) and community workers | TS—OHNP | Zoonoses commission | TS-OHNP/TFP/Sectoral |
| | | 2.2 | Distribute the tablets | Authorities (animal health and environment) and community workers | TS—OHNP | Zoonoses commission | TS-OHNP/TFP/Sectoral |
| 3 | Agents have notification forms (paper or electronic) and are trained in their use | 3.1 | Develop notification forms (paper and electronic) for slaughterhouses and slaughter areas | Abattoirs, ZATE, Areas for abattoir slaughter, Nurses | Zoonoses Commission | Technical directorates | TS-OHNP/TFP/Sectoral |
| | | 3.2 | Update sector-specific fact sheets in electronic format for animal, human and environmental health for harmonization | Service points of the various ministries | Zoonoses Commission | Technical directorates | TS-OHNP/TFP/Sectoral |
| | | 3.3 | Organize training on the use of tablets and electronic notification forms | Authorities (animal health and environment) and community workers | TS—OHNP | TS—OHNP; technical directorates | TS-OHNP/TFP/Sectoral |
| 4. | Posters on case definition are displayed in veterinary posts, health centres and eco-guards' offices. | 4.1 | Create poster support | Service points of the various ministries | TS—OHNP | Zoonoses commission, with communication support | TS-OHNP/TFP/Sectoral |
| | | 4.2 | Print and distribute posters | Service points of the various ministries | TS—OHNP | Zoonoses commission, with communication support | TS-OHNP/TFP/Sectoral |
| 5. | Agents are trained in anthrax epidemiology and the recognition of suspected cases | 5.1 | Identify and train epidemiology trainers | Authorities (animal health and environment) and community workers | TS—OHNP | TS—OHNP; technical directorates | TS-OHNP/TFP/Sectoral |
| | | 5.2 | Provide cross-sectoral training for trainers in epidemiology and the recognition of suspected cases | Regional level | Zoonoses commission | National experts in anthrax and epidemiology | TS-OHNP/TFP/Sectoral |
| | | 5.3 | Provide training for agents, adapted to the sector and administrative levels of the beneficiaries, with a common core | Provincial, local level | Zoonoses commission | Agents trained by national anthrax experts | TS-OHNP/TFP/Sectoral |

*(Continued)*

**Table 2.** (Continued)

| CHANGES | | ACTION | | ACTION BENEFICIARIES | MANAGER | EXECUTANT | ORIGIN OF FUNDS |
|---|---|---|---|---|---|---|---|
| 6. | The trackers and eco-guards have image boxes that allow them to recognize cases | 6.1 | Update the image box or create a new box | Eco-guards, trackers | Environment technical directorate | Head of surveillance activities | TS-OHNP/ TFP/ Sectoral |
| | | 6.2 | Disseminate image boxes based on existing unusual events to all eco-guards and trackers | Eco-guards, trackers | Environment technical directorate | Head of surveillance activities | TS-OHNP/ TFP/ Sectoral |
| 7 | Community leaders are trained and involved in surveillance activities | 7.1 | Identify the community leaders who will be involved | Service points of the various ministries, community leaders | Agents of service points, managers of private human or animal health structures | Agents of services points (CSPS, ZATE, PV, PF), managers of private health structures (clinic, human or animal care practice) | TS-OHNP/ TFP |
| | | 7.2 | Plan meetings with community leaders | Service points of the various ministries, community leaders | Agents of service points, managers of private human or animal health structures | Agents of services posts (CSPS, ZATE, PV, PF), managers of private health structures (clinic, human or animal care practice) | TS-OHNP/ TFP |
| | | 7.3 | Conduct leaders' courtesy visit with multi-sector teams | Service points of the various ministries, community leaders | Agents of service points, managers of private human or animal health structures | Agents of services posts (CSPS, ZATE, PV, PF), Managers of private health structures (clinic, human or animal care practice) | TS-OHNP/ TFP |
| | | 7.4 | Sensitize leaders to anthrax and how to notify with appropriate tools (photos, sketches) | Population, leaders, service points of various ministries, community leaders | Zoonosis and surveillance commissions | Agents of services posts (CSPS, ZATE, PV, PF), managers of private health structures (clinic, human or animal care practice) | TS-OHNP/ TFP |
| 8 | People are aware of the risk of anthrax and the importance of notifying suspected cases and are trained to do it | 8.1 | Train actors on the importance of joint communication | Service points of various ministries, private health care workers | Zoonosis and surveillance commissions | OHNP—Decentralized structures | TS-OHNP/ TFP |
| | | 8.2 | Harmonizing communication tools | Service points of various ministries, private health care workers | Zoonosis and surveillance commissions | OHNP—Decentralized structures | TS-OHNP/ TFP |
| | | 8.3 | Develop a joint information and awareness program | Service points of various ministries, private health care workers | Zoonosis and surveillance commissions | OHNP—Decentralized structures | TS-OHNP/ TFP |
| | | 8.4 | Planning the various training courses | Service points of various ministries, private health care workers | Zoonosis and surveillance commissions | Zoonosis and surveillance commissions | TS-OHNP/ TFP |
| | | 8.5 | Organize joint programmes (TV, radio) | Service points of various ministries, private health care workers | Zoonosis and surveillance commissions | OHNP—Decentralized structures | TS-OHNP/ TFP |
| | | 8.6 | Organize joint anthrax awareness campaigns by the various sectors | Service points of various ministries, private health care workers | Zoonosis and surveillance commissions | OHNP—Decentralized structures | TS-OHNP/ TFP |
| 9 | Butchers are trained to recognise anthrax lesions | 9.1 | Identify butchers | Butchers, slaughterhouse and slaughter area inspectors, ministry service points | Provincial and regional directorates (in charge of inspections) | Provincial and regional directorates (in charge of inspections) | Sectoral/ TFP |
| | | 9.2 | Plan training sessions | Butchers, abattoir and slaughter area inspectors, ministry service points | Provincial and regional directorates (in charge of inspections) | Provincial and regional directorates (in charge of inspections) | Sectoral/ TFP |

(*Continued*)

**Table 2.** (Continued)

| CHANGES | | ACTION | | ACTION BENEFICIARIES | MANAGER | EXECUTANT | ORIGIN OF FUNDS |
|---|---|---|---|---|---|---|---|
| 10 | Agents are assigned and empowered according to their motivation | 10.1 | Evaluate staffing needs by zone | Human resources directorates of the ministries concerned, key surveillance actors | Regional directorates (environment, animal, and human health) | Regional and human resources directorates | Sectoral/TFP |
| | | 10.2 | Feedback evaluation results to stakeholders | Human resources directorates of the ministries concerned, key surveillance actors | Regional directorates (environment, animal, and human health) | Regional and human resources directorates | Sectoral/TFP |
| | | 10.3 | Create performance indicators | Human resources directorates of the ministries concerned, key surveillance actors | Regional directorates (environment, animal, and human health) | Regional and human resources directorates | Sectoral/TFP |
| 11 | There is effective leadership in the conduct of surveillance activities | 11.1 | Create concertation frameworks between stakeholders in the ministries concerned | Key surveillance actors | TS—OHNP | OHNP—Decentralized structures | TS-OHNP/TFP |
| | | 11.2 | Draft surveillance regulations based on the OH concept | Ministries concerned | TS—OHNP | Zoonoses and surveillance commissions | TS-OHNP/TFP |
| 12 | Actions are implemented to motivate agents | 12.1 | Draw up a motivation grid | Key surveillance actors | TS—OHNP | Zoonoses and surveillance commissions | TS-OHNP/TFP |
| | | 12.2 | Send letters of congratulations and awards to deserving actors | Key surveillance actors | TS—OHNP and central directorates of the ministries concerned | Central and regional directorates of the ministries concerned | Sectoral |
| 13 | Existing paper and electronic notices are simplified | 13.1 | Organize a workshop to update data collection sheets | Service points of various ministries, private health care workers | TS—OHNP and central directorates of the ministries concerned | Zoonoses and surveillance commission | TFP |
| | | 13.2 | Create an electronic form | Service points of various ministries, private health care workers | Zoonosis and surveillance commissions | Information technology directorates of the ministries concerned | TFP |
| | | 13.3 | Develop an easy-to-use application for filling in electronic forms | Service points of various ministries, private health care workers | Zoonosis and surveillance commissions | Information technology directorates of the ministries concerned | TFP |

CSPS: Centre de Santé et de Promotion Sociale (health and social promotion centre; PF: Poste forestier (forestry post); OHNP: One Health National Platform; SS: Sanitary services; TFP: Technical and financial partners; TS: Technical secretary; PV: Poste vétérinaire (Veterinary post); ZATE: Zone d'Appui Technique à l'Elevage (Livestock Technical Support Zone).

however, were unfamiliar with the concept, and interpreted integrated surveillance as meaning good coordination and collaboration between actors within the same intra-sectoral surveillance programme, operating at different decision-making levels. The definition finally agreed upon by the participants is consistent with those in the literature for integrated surveillance in a One Health approach [13,17]. The co-construction process, by providing a framework for participants to compare their perspectives, highlighted several points of divergence but also enabled participants to engage in a learning process towards greater mutual understanding.

The quality of a participatory process significantly impacts the quality of its results [18]. The ability of any participatory process to produce relevant results is based on several elements: (i) the representativeness, knowledge and influence of the participants; (ii) the ability of the method to elicit, compile and synthesize information from different sources to create representations to which participants can reasonably adhere; and (iii) the quality of the facilitation

[15]. The results produced by the participatory process must therefore be analysed in the light of the quality of the process itself.

Despite our efforts to bring together a panel of participants who are representative of the key actors in the anthrax surveillance system in Burkina Faso, it was difficult to benefit from the presence of certain actors and/or maintain their commitment throughout the workshop. Moreover, it was not just a question of having all categories of stakeholders represented, but also of the representative sent by the invited institution being relevant to the workshop's objective. This was not the case for the representative of the animal health sector at local level, who had neither a role in nor experience of anthrax surveillance. There is also the question of how representative a participant's point of view is of the institution represented. Indeed, points of view can diverge from one individual to another within the same category of stakeholders and inviting a single representative from a given institution to this type of process is likely to be insufficient to capture and consider exhaustively the knowledge and expectations of that category [9,19]. Finally, the ability of the process to generate change outside its arena is strongly linked to the participants' power in influencing decision-making. In our case, where the main objective was to produce an action plan, the question of its implementation arises if the collective decisions taken during the workshop are not taken to a high enough decision-making level [8]. Some categories of stakeholders appeared to be more involved than others in the issue of One Health surveillance. Environmental and animal health participants showed a greater interest than their human health counterparts. There are two possible reasons for this. The first is that One Health surveillance is still viewed in a very anthropocentric way. In this scenario, the workshop thus represented an opportunity for the animal and environmental sectors, which have more limited resources for surveillance than the human sector, to argue in favour of a better distribution of available resources between sectors, and also between levels of intervention [8]. The second is that zoonoses remain a marginal issue for the human health sector, particularly compared with infectious diseases of non-animal origin (such as malaria or HIV) and non-infectious diseases (such as diabetes), which represent a greater health and socio-economic impact [20]. But ensuring that stakeholders are properly represented is not enough to guarantee the success of a participatory process. The challenge then lies in ensuring that a plurality of viewpoints is expressed and combined to generate new knowledge and collective decisions. Managing diversity can be complex, posing implementation difficulties that compromise the achievement of the objectives sought through the process [21]. Furthermore, participatory processes require more commitment and energy from participants than traditional approaches, where participants are more passive [22]. The quality of the methodology adopted and the ability of the facilitation team to implement it are therefore key determinants of the success of a participatory process [15,18]. The composition of the facilitation team and the quality of facilitation are also critical to the success of such a process, as the facilitator or facilitators hold the keys to the change process [11]. On the one hand, it is important to have people in that role who will moderate in an almost "naïve" way and not bring their own opinion to bear on what participants say. On the other hand, it can be beneficial for facilitators to have some expertise in the field, to enable them to challenge participants on potential inconsistencies or areas that may not have been addressed [15]. Indeed, it is not certain that the mere confrontation of different points of view will lead to an accurate analysis of the situation and needs, let alone to a compromise. The facilitator therefore also has a role to play in sharing and debating objective information [11]. In addition, the facilitator's role is to ensure that the plurality of viewpoints is considered. To do this, the facilitator must be able to manage conflicts and power dynamics that may represent an obstacle to the collective process [23]. For our workshop, the facilitation team was diverse, including contextual, technical, and methodological experts.

Despite all the efforts to develop participatory processes that are inclusive, adapted to the context and adaptive, participatory processes are complex and influenced by social and political factors that are beyond the control of the process initiators [10].

The power relationships between institutions and stakeholder categories outside the participatory forum are likely to be reproduced within the process, and some participants are likely to have a greater influence than others on the decisions taken. There is therefore a risk that the participatory process may actually reinforce power imbalances by presenting its results as the fruit of a collective reflection, while the contributions from actors of different categories may have been unequal and the results may in fact mainly reflect the concerns of the actors from dominant categories [24]. During our workshop, for example, the hierarchical position and high social recognition of some participants sometimes hampered the free debate of ideas and sharing of perspectives. Moreover, the act alone of expressing their point of view publicly does not guarantee that stakeholders from categories with less power will truly be listened to and heard. Indeed, where there are tensions, free expression by participants can even have the opposite effect, by reinforcing those points of tension and power imbalances [11].

In the same vein, the decisions that will be taken, even in a collective and concerted manner, as part of the participatory process, have no certainty of success because they are dependent on elements external to the process [12]. In the case of our participatory process, most proposed actions are dependent on the availability of a sufficient budget, itself largely dependent on funding granted by technical and financial partners, who may have different priorities from those that guided the action plan. Implementation of the action plan co-constructed with the stakeholders therefore lies not so much in the changes engendered in the surveillance stakeholders as in changes in the rules of the game that determine decision-making and financial flows [11].

Finally, despite the quality of the method and facilitation, in some cases it may be difficult to engage certain individuals who have cultural values and cognitive framework that hinder their participation in this type of co-construction process that is based on listening, sharing, and co-learning [10].

The co-construction process implemented in Burkina Faso took place over a three-day workshop. It would have benefited from being more iterative, through the organization of further workshops during which participants could have reflected on the results produced, to discuss and develop them further [14,25]. This would also have been an opportunity to usefully involve certain categories of stakeholders not included during this first one (farmers, veterinarians and medical practitioners). Moreover, in the case of anthrax surveillance, the front-line actors are those working at local level, and it would have been interesting to organize workshops with a wider range of local actors by decentralizing them to a sub-national level. In addition, it would be interesting to put a figure on the cost of the actions identified to facilitate the search for funding.

The co-construction process proposed in Burkina Faso to collectively define the modalities for implementing the One Health surveillance system for anthrax appears satisfactory in terms of the results produced and the active participation of those invited. Participants expressed their satisfaction regarding the workshop organization and the results produced, and stated that they would take actions to contribute to the application of the decisions taken. They also considered that they improved their knowledge of the epidemiology of anthrax, as well as of the organization and functioning of anthrax surveillance. However, this evaluation remains highly subjective. Despite the anticipated benefits, participatory processes are still largely controversial. They are often criticized by decision-makers as costly in terms of time and resources [25]. They are often questioned because the results produced are very largely dependent on the quality of facilitation and the representativeness of participants [14]. Additionally, from our

operational perspective, the results produced cannot claim to be the most suitable and appropriate for the simple reason that they were co-constructed, since they are representative of only part of the viewpoints and were influenced by the social and political context in which they were expressed [10]. It is therefore necessary to explicitly document the implementation of the process, to assess its intrinsic quality (method used, facilitation, representativeness of participants) and to measure its effects (changes engendered among participants, or on the implementation context) in a robust way, while considering other factors that could also have contributed to the observed effects [13]. Monitoring and evaluating the participatory process is also an opportunity to highlight elements that may impact the quality and effects of the process and therefore redirect the course of the process to take these elements into account [12].

## Conclusions

The process of co-constructing a One Health surveillance system for anthrax in Burkina Faso enabled representatives of surveillance stakeholders to collectively define their vision of the ideal One Health surveillance system for anthrax. They were then able to establish the path to be taken from the current situation to this desired vision of anthrax surveillance. This co-construction process was also an opportunity for the participants to engage in a process of technical and social learning, by exchanging their points of view on the problem and the solutions for dealing with it. This pooling of knowledge, which generates new collective knowledge, creates a conducive environment for making collective decisions and planning future action. The interactions that take place between participants during the process provide the basis for new partnerships and collaboration, conducive to the implementation of more integrated health policies.

The methodological framework we developed and used to conduct the participatory process consists in four key steps: defining the ideal situation or vision of the future; characterizing the current situation; identifying changes to achieve the ideal situation; and drafting an action plan to operationalize the changes. However, the implementation of the method needs to be adaptive and iterative, adapting along the way not only to the knowledge shared by the participants, but also to the posture of the participants in relation to the approach and to each other. The adopted method and the quality of facilitation is therefore a key element in the success of the participatory process.

Each participatory process is unique because it is specific to the context in which it is implemented, the participants mobilized, and the objectives pursued [12]. However, our study proposes a methodological framework that can be used, after adaptation, in other contexts to address different issues.

## Supporting information

**S1 File. Inclusivity in global research questionnaire.**
(PDF)

## Acknowledgments

The authors would like to thank all the participants who agreed to take part in the participatory process.

## Author Contributions

**Conceptualization:** Sougrenoma Désiré Nana, Potiandi Serge Diagbouga, Pascal Hendrikx, Marion Bordier.

**Formal analysis:** Sougrenoma Désiré Nana, Marion Bordier.

**Funding acquisition:** Pascal Hendrikx, Marion Bordier.

**Investigation:** Sougrenoma Désiré Nana, Raphaël Duboz, Marion Bordier.

**Methodology:** Sougrenoma Désiré Nana, Raphaël Duboz, Pascal Hendrikx, Marion Bordier.

**Supervision:** Potiandi Serge Diagbouga, Pascal Hendrikx, Marion Bordier.

**Writing – original draft:** Sougrenoma Désiré Nana.

**Writing – review & editing:** Raphaël Duboz, Potiandi Serge Diagbouga, Pascal Hendrikx, Marion Bordier.

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
