## [Decision Letter · Decision Letter 0]

27 Feb 2024

PONE-D-23-41971A participatory approach to support the development of a One Health surveillance system for anthrax in Burkina FasoPLOS ONE

Dear Dr. Bordier,

Thank you for submitting your manuscript to PLOS ONE. After careful consideration, we feel that it has merit but does not fully meet PLOS ONE’s publication criteria as it currently stands. Therefore, we invite you to submit a revised version of the manuscript that addresses the points raised during the review process.

We look forward to receiving your revised manuscript.

Kind regards,

Chisoni Mumba

Academic Editor

PLOS ONE

Journal Requirements:

This work was funded in part by CIRAD, VetagroSup (the French institute for higher education and research in food, animal health, agricultural and environmental sciences) and the French Embassy in Burkina Faso.

Additional Editor Comments:

Dear Authors

Please attend to the comments by both Reviewer 1 and 2. Please pay attention to the tracked changes in the manucript by Reviewer number 2 and attend to them line by line.

Reviewers' comments:

Reviewer's Responses to Questions

**Comments to the Author**

1. Is the manuscript technically sound, and do the data support the conclusions?

Reviewer #1: Yes

Reviewer #2: Yes

2. Has the statistical analysis been performed appropriately and rigorously? 

Reviewer #1: N/A

Reviewer #2: N/A

3. Have the authors made all data underlying the findings in their manuscript fully available?

Reviewer #1: No

Reviewer #2: Yes

4. Is the manuscript presented in an intelligible fashion and written in standard English?

Reviewer #1: Yes

Reviewer #2: Yes

5. Review Comments to the Author

Reviewer #1: This manuscript reports the results of a co-construction approach to develop a One Health surveillance system for anthrax in Burkina Faso. Some actions are proposed by the participants to improve the current surveillance system.

Albeit some minor mistakes, the manuscript is well written. The methods are appropriate for the objectives and the conclusion is supported by the result. I provide here comments for the author`s consideration.

Introduction

The authors don’t mention the presence of a One Health National Platform in Burkina Faso. So, among the participant invited, this stakeholder is don't take account. Even if another paper is already published on the current involved stakeholders in the anthrax surveillance, it is necessary to remind briefly here.

Material and methods:

In the participatory process, it is necessary to implement the system and use results to correct the system. That can constitute the fifth step.

When you say Central and local authorities, it is necessary to precise the level of the positioning (National, provincial, departmental, village, etc.). In the table 2, at the points 5.2 & 5.3, you mentioned Regional, provincial and local level. What “local level” means?

For each institution, what kind of personal (head or agent) was took account? That is important in the future implementation of the changes’ actions.

From the Table 1, can you explain the reason of the total or partial absence of some stakeholder? What were the financial supported for the participants? That can explain also the quality of the participants.

Lines 98 – 99: “appropriate participative tools” used can be cited here. Tools used are so important as the quality of facilitators.

Lines 104 – 106: The evaluation is an essential part of the process building. The evaluation method must be presented in this paper.

Figure 2: Why different boxes has the same number? The numbers must be different and classified.

Review also the abscissa axis texts: What mean T0? T0 + 10 ans?

RESULTS

Line 181: notification of suspected or confirmed cases? Need the precision. In the lines 201 to 207, it is mentioned the suspected cases.

How, the time 10 years had been defined? Is it the time of evaluation of the impact?

Figure 3, 4, 5, 6 & 7. Review pictures’ resolutions for a best reading.

Table 2: Repeat the columns’ titles in each page can help for the best reading du to the width of this table.

In the point 7 of the table 2, keep the original denomination of all services: ZATE, CSPS, PV, PF, etc.)

The action plan presented in the table 2 is not planed in time. How that can be reached before the 10 years?

At this stage of your study, is it possible to know if process succeed or not? That is not clear in the manuscript.

Discussion

Is it possible to evaluate the cost of the changes proposed to improve the current surveillance system? Also discuss the feasibility of this changes regarding the cost, and the financial sources.

Reviewer #2: The co-authors shall address comments provided in the attachment and I can review the revised manuscript.

The co-authors should make it very clear the fact that the proposed One Health anthrax surveillance system is developed by creating bridges between existing sector specific surveillance system.

6. PLOS authors have the option to publish the peer review history of their article (what does this mean?). If published, this will include your full peer review and any attached files.

Reviewer #1: **Yes: **Sié Hermann Pooda

Reviewer #2: **Yes: **Serge Nzietchueng

---

## [Author Response · Author response to Decision Letter 0]

5 Apr 2024

All responses to editor and reviewers have been submitted through the "cover letter file" and the "response to reviewers" file.

---

## [Decision Letter · Decision Letter 1]

21 May 2024

A participatory approach to move towards a One Health surveillance system for anthrax in Burkina Faso

PONE-D-23-41971R1

Dear Dr. Bordier,

We’re pleased to inform you that your manuscript has been judged scientifically suitable for publication and will be formally accepted for publication once it meets all outstanding technical requirements.

Kind regards,

Chisoni Mumba

Academic Editor

PLOS ONE

Additional Editor Comments (optional):

Reviewers' comments:

Reviewer's Responses to Questions

**Comments to the Author**

1. If the authors have adequately addressed your comments raised in a previous round of review and you feel that this manuscript is now acceptable for publication, you may indicate that here to bypass the “Comments to the Author” section, enter your conflict of interest statement in the “Confidential to Editor” section, and submit your "Accept" recommendation.

Reviewer #2: All comments have been addressed

2. Is the manuscript technically sound, and do the data support the conclusions?

Reviewer #2: Yes

3. Has the statistical analysis been performed appropriately and rigorously? 

Reviewer #2: No

4. Have the authors made all data underlying the findings in their manuscript fully available?

Reviewer #2: Yes

5. Is the manuscript presented in an intelligible fashion and written in standard English?

Reviewer #2: Yes

6. Review Comments to the Author

Reviewer #2: The previous comments have been addressed and the revised manuscript is better.

Accepted with minor review. See comments provided for consideration, No need to resubmit for a third review

7. PLOS authors have the option to publish the peer review history of their article (what does this mean?). If published, this will include your full peer review and any attached files.

Reviewer #2: **Yes: **Serge Nzietchueng

---

## [Editor Report · Acceptance letter]

27 May 2024

PONE-D-23-41971R1 

PLOS ONE

Dear Dr. Bordier, 

I'm pleased to inform you that your manuscript has been deemed suitable for publication in PLOS ONE. Congratulations! Your manuscript is now being handed over to our production team.

Kind regards, 

on behalf of

Dr Chisoni Mumba 

Academic Editor

PLOS ONE